# Gastrodin and Gastrodigenin Improve Energy Metabolism Disorders and Mitochondrial Dysfunction to Antagonize Vascular Dementia

**DOI:** 10.3390/molecules28062598

**Published:** 2023-03-13

**Authors:** Sha Wu, Rong Huang, Ruiqin Zhang, Chuang Xiao, Lueli Wang, Min Luo, Na Song, Jie Zhang, Fang Yang, Xuan Liu, Weimin Yang

**Affiliations:** 1Yunnan Key Laboratory of Pharmacology for Natural Products, Kunming Medical University, Kunming 650500, China; 2Shanghai Frontiers Science Center of TCM Chemical Biology, Institute of Interdisciplinary Integrative Medicine Research, Shanghai University of Traditional Chinese Medicine, Shanghai 201203, China; 3School of Basic Medicine, Kunming Medical University, Kunming 650500, China

**Keywords:** vascular dementia, gastrodin, gastrodigenin, pharmacological action, energy metabolism, mitochondrial function

## Abstract

Vascular dementia (VD) is the second most common dementia syndrome worldwide, and effective treatments are lacking. *Gastrodia elata* Blume (GEB) has been used in traditional Chinese herbal medicine for centuries to treat cognitive impairment, ischemic stroke, epilepsy, and dizziness. Gastrodin (*p*-hydroxymethylphenyl-b-*D*-glucopyranoside, Gas) and Gastrodigenin (*p*-hydroxybenzyl alcohol, HBA) are the main bioactive components of GEB. This study explored the effects of Gas and HBA on cognitive dysfunction in VD and their possible molecular mechanisms. The VD model was established by bilateral common carotid artery ligation (2-vessel occlusion, 2-VO) combined with an intraperitoneal injection of sodium nitroprusside solution. One week after modeling, Gas (25 and 50 mg/kg, i.g.) and HBA (25 and 50 mg/kg, i.g.) were administered orally for four weeks, and the efficacy was evaluated. A Morris water maze test and passive avoidance test were used to observe their cognitive function, and H&E staining and Nissl staining were used to observe the neuronal morphological changes; the expressions of Aβ1-42 and p-tau396 were detected by immunohistochemistry, and the changes in energy metabolism in the brain tissue of VD rats were analyzed by targeted quantitative metabolomics. Finally, a Hippocampus XF analyzer measured mitochondrial respiration in H_2_O_2_-treated HT-22 cells. Our study showed that Gas and HBA attenuated learning memory dysfunction and neuronal damage and reduced the accumulation of Aβ1-42, P-Tau396, and P-Tau217 proteins in the brain tissue. Furthermore, Gas and HBA improved energy metabolism disorders in rats, involving metabolic pathways such as glycolysis, tricarboxylic acid cycle, and the pentose phosphate pathway, and reducing oxidative damage-induced cellular mitochondrial dysfunction. The above results indicated that Gas and HBA may exert neuroprotective effects on VD by regulating energy metabolism and mitochondrial function.

## 1. Introduction

Vascular dementia (VD) is a syndrome of severe cognitive impairment caused by vascular risk factors, the incidence of which increases with age [1]. Risk factors include atherosclerosis, thrombosis, or other vascular lesions [2]. In addition, VD pathology is usually accompanied by cerebral microinfarction, vascular damage, and neuronal atrophy, except for Alzheimer’s disease (AD)-like lesions [3]. Current studies on the pathogenesis of VD include neuronal loss [4], energy metabolism [5], mitochondrial dysfunction [6], glutamate neurotoxicity [7], and toxic substance accumulation [8]. Among these, energy metabolism and mitochondrial dysfunction have raised widespread concern because the brain, as the most active organ of energy metabolism, requires a continuous supply of energy from ATP. When VD occurs, cerebral ischemia and hypoxia can produce various toxic substances that impair mitochondrial homeostasis and energy production, and the brain’s high demand for energy can lead to the further development of dementia [9]. Normal mitochondrial homeostasis plays a vital role in stabilizing energy metabolism in the brain [10], although commonly used dementia drugs can show cognitive benefits in some patients with VD. However, few patients achieve the desired therapeutic outcomes [11]. The rhizome of *Gastrodia elata* Blume (GEB) is a Chinese herbal medicine that has the effects of regulating the circulatory system, sedation, anti-epilepsy, anti-convulsion, anti-depression, anti-inflammation, anti-oxidation, improving memory, and anti-aging [12]. Gastrodin (*p*-hydroxymethylphenyl-b-*D*-glucopyranoside, Gas) and Gastrodigenin (*p*-hydroxybenzyl alcohol, HBA) are active components of the rhizome of Gastrodia elata and are of interest for their various pharmacological activities in the central nervous system. Previous studies have reported that VD rats can cause cognitive impairment and VD-like pathogenesis characterized by Aβ and Tau deposition in the hippocampus [13,14]. Gas improves cognitive deficits in VD rats by reducing toxic substance accumulation (Aβ and Tau proteins) [15] and by reducing excessive autophagy and apoptosis of neuronal cells [16]. After entering the central nervous system, Gas is metabolized to HBA [17]. Previously, HBA treatment was reported to prevent hypomnesis [18] and significantly inhibit oxidative stress and excitotoxicity to suppress neuronal death [19,20].

Despite solid evidence of their beneficial effects against nervous system diseases, studies of Gas and HBA on energy metabolism and mitochondrial function in VD brain tissue are limited. This study aimed to investigate the pharmacological effects of Gas and HBA on VD and the molecular mechanisms. The VD model was established by bilateral common carotid artery occlusion (2-VO), characterized by cerebral ischemia and hypoxia-induced by chronic hypoperfusion and can better simulate human VD caused by atherosclerosis and arterial lumen stenosis. Bilateral common carotid artery ligation (2-vessel occlusion, 2-VO) is characterized by chronic hypoperfusion-induced cerebral ischemia and hypoxia, which better simulates VD in humans due to atherosclerosis and arterial lumen narrowing [21]. In the context of the beneficial intervention of Gas and HBA, the molecular mechanism of the improvement of VD in mitochondria and bioenergetics was proposed.

## 2. Results

### 2.1. Gas and HBA Improve Behavioral and Cognitive Alterations in VD Rats

We observed apparent local atrophy and necrosis in the brain tissue of VD rats, and the morphology was improved after the intervention of Gas and HBA (Figure 1A). A Morris water maze (MWM) test was used to evaluate the spatial learning memory function after Gas and HBA treatment. The localization navigation test lasted five days, and the escape latency of animals in the Model group was significantly longer than in the Sham group, while the MWM test was significantly relieved by Gas and HBA treatment (Figure 1B). Subsequently, in the probe trial on day 7, a long time was spent in the platform and target quadrant, and more search times were observed in the Gas and HBA group compared to the Model group (Figure 1C–F). No difference was observed in the swimming speed covered among all groups indicating normal motor functions (Figure 1G) and a representative picture of the swimming tracks (Figure 1H). Subsequently, a passive avoidance test was performed to examine the learning and short-term memory abilities of the Model rats. The step-through latency of the rats in the Gas and HBA group was significantly higher than that of the Model group (Figure 1I), while there was no significant difference in the electric time and the number of errors (Figure 1J,K). In conclusion, Gas and HBA can prevent cognitive decline in VD rats.

### 2.2. Gas and HBA Reduce Attenuates 2-VO-Induced Neuronal Damage in the Hippocampus

The results of H&E staining in VD rats’ hippocampi showed neuronal disturbances in CA1 and CA3 regions, including loss, degeneration, pyknotic nuclei, and severe cellular edema. Notably, these pathological characteristics were attenuated following Gas and HBA administration, especially in the high dose (50 mg/kg) (Figure 2A,C). We subsequently examined the neuronal integrity by Nissl staining (Figure 2B,D), and the results showed that the VD rats had significantly reduced neuron function with a few Nissl bodies (blue). In contrast, Gas and HBA treatment significantly attenuated the neuronal loss. These data suggest that Gas and HBA treatment mitigates neuronal damage in VD rats.

### 2.3. Gas and HBA Reduce Aβ and Tau Protein Expression in the Brain of VD Rats

2-VO induces Aβ and Tau protein deposition [22,23], and our previous study [15] showed that Gas ameliorates cognitive deficits in VD rats by inhibiting abnormal phosphorylation of Aβ and Tau. Thus, we found that Gas and HBA reduced the protein levels of Aβ1-42 (Figure 3A,C), p-tau396 (Figure 3B,D), and p-tau217 (Figure 3E) in the brain tissue of VD rats.

### 2.4. Gas and HBA Inhibit the Changes in Energy Metabolism in VD Rats

The above results show that 2-VO can lead to significant cognitive impairment in rats, while the intervention with Gas and HBA can effectively improve cognitive function. It has been reported that normal energy metabolism is closely related to common cognitive responses, and therefore any defect in energy metabolic processes can lead to a decrease in cognitive function [24]. With the premise that Gas and HBA interventions are effective, we quantified 40 metabolites that target brain energy metabolism (glycolysis, TCA cycle, and pentose phosphate pathway) (Appendix A). In the Sham, Model, HBA-50, and Gas-50 groups, principal component analysis (PCA) characterized a distinct trend of separation of metabolic profiles among the groups, suggesting differences in metabolic patterns (Figure 4A). The heat map was used to visualize the metabolite profiles among the groups (Figure 4B). Orthogonal partial least squares discriminant analysis (OPLS-DA) and variable importance projection (VIP) were used to screen for differential metabolites between groups with screening criteria: VIP > 1 and *p* ≤ 0.05. There were 16 differential metabolites in the Sham vs. Model group (Figure 4C), of which five were downregulated and 11 were upregulated. Among the metabolites that were abnormally elevated or decreased in the Mode group, 13 metabolite levels were significantly back-regulated after Gas and HBA intervention. In the Gas group, seven were upregulated and six were downregulated, respectively (Figure 4D). In addition, eight metabolite levels were significantly regulated after HBA intervention, four upregulated and four down-regulated (Figure 4E).

### 2.5. Gas and HBA Protect HT-22 Cells from H_2_O_2_-Induced Damage

As shown, treatment of HT-22 cells with concentrations ranging from 12.5–100 μM Gas and HBA had no significant effect on cell viability (Figure 5A,B). Using 500 μM H_2_O_2_ treatment of HT-22 cells for 12 h caused oxidative damage model (Figure 5C,D). Treatment with 12.5–100 μM Gas and HBA significantly improved the H_2_O_2_-reduced HT-22 cell viability (Figure 5E,F).

### 2.6. Gas and HBA Attenuated Mitochondrial Dysfunction in HT-22 Cells Induced by H_2_O_2_

Previous energy metabolomics analysis suggested changes in brain tissue energy metabolism in VD rats; we therefore determined the mitochondrial respiratory efficacy of H_2_O_2_-treated HT-22 cells. Mitochondrial respiratory efficacy was significantly reduced in H_2_O_2_-treated cells, whereas Gas and HBA pretreatment partially mitigated the H_2_O_2_-induced inhibition of mitochondrial respiration in HT-22 cells (Figure 6A). Gas and HBA pretreatment significantly countered the H_2_O_2_-induced reduction in basal respiration (Figure 6B), maximal respiration (Figure 6C), and ATP production (Figure 6D) in HT-22 cells. These results suggest that Gas and HBA partially attenuated H_2_O_2_-induced mitochondrial dysfunction in HT-22 cells.

Gas and HBA may exert neuroprotective effects by antagonizing VD through ameliorating energy metabolism disorders in VD rats and attenuating oxidative damage-induced cellular mitochondrial dysfunction (Figure 7).

## 3. Material and Method

### 3.1. Chemicals and Materials

Gas (C_13_H_18_O_7_; molecular weight: 286.28; purity ≥ 98%) and HBA (C_13_H_18_O_7_; molecular weight: 286.28; purity ≥ 98%) were purchased from Nanjing Zelang Medical Technology Co. Gas, and HBA was dissolved in normal saline (NS). All reagents were of analytical reagent grade. Oligomycin, FCCP (2-[2-[4-(trifluoromethoxy)phenyl]hydrazinylidene]-propanedinitrile), and rotenone were purchased from Sigma-Aldrich (St. Louis, MO, USA). Anti-beta Amyloid (1:200, Proteintech, Wuhan, China) and P-Tau 396 (1:100, Abcam, Shanghai, China). 

### 3.2. Animals

Adult male Sprague–Dawley (SD) rats weighing 270 ± 10 g were purchased from the Experimental Animal Center of Kunming Medical University. Before the experiment, all animals were allowed adaptive feeding for a week. The rats were kept under a 12 h light/dark cycle at 22–24 °C with free access to food and water. The animals were divided into the following six groups: control group (Sham), 2-VO group (Model), Gas 25 mg/kg group (Gas-25), Gas 50 mg/kg group (Gas-50), HBA 25 mg/kg group (HBA-25), and HBA 50 mg/kg group (HBA-50). All animals were given Gas, HBA, or NS orally on day 7 after surgery, daily for 28 days. The progress of the experiment is shown in Figure 1.

### 3.3. VD Rats Model Preparation

2-VO prepared the animal model of VD. SD rats were continuously anesthetized with 200 g, 0.2 mL/min of 2% isoflurane gas. After disinfection with 75% ethanol, the skin was cut in the middle of the neck, and the subcutaneous tissue was bluntly separated. The common carotid artery pulsated at the angle between the trapezius muscle and the trachea exactly below the incision and was then ligated and severed with surgical suture. Control rats underwent the same surgical procedure, with the exception that the common carotid artery was exposed but not ligated. Aseptic operation was kept during the operation. After the operation, antibiotics were dropped to prevent infection and then sutured, and sodium nitroprusside injection was injected into the abdominal cavity. The intraoperative temperature was about 37 °C until the rats woke up.

### 3.4. Morris Water Maze Test

The Morris water maze (Shanghai Xin Luan MDT infotech LTD., Shanghai, China) test assessed spatial learning and memory and was carried out as previously described. The MWM consists of a large round pool (120 cm in diameter and 50 cm in height) filled with white non-toxic powder. The pool was divided equally into four quadrants, with a 20 cm-diameter round platform hidden in the center of the target quadrant. Before the experiment, the rats were adaptively trained for one day; the positioning navigation experiment then lasted for five days, and the system automatically recorded the track of the rats as they were placed into the water from the edge of the pool. Rats who failed to find the platform within the 90 s would be guided to the platform and allowed to stay there for 10 s. For the probe trial, the platform needs to be removed, and the track of the rat within the 90 s is recorded.

### 3.5. Passive Avoidance Test

The cognitive abilities of rats were tested using a SUPERAS shuttle box (Shanghai Xin Luan MDT infotech LTD., Shanghai, China). The test equipment consisted of a light and dark box and an electrical stimulation controller for three days of testing. On the first day, the rats were placed in the light box, and after 30 s, the door between the light and dark boxes was opened. Rats have solid exploratory behaviour and prefer darkness to light. Therefore, the rats would enter the dark box quickly; once they were fully inside, we immediately shut the door and they were given an electric shock. The intensity of the electric shock was the minimum current (0.6 mA for 10 s) that could cause the rat to flinch and vocalize. Having allowed the rats to remain in the dark box for 30 s (to allow the animals to form an association between the box and the electric shock), we placed them back in the cage. On the second day, the rats moved freely in the shuttle box for 5 min, and the electrical stimulator would automatically activate after entering the dark box. Day 3 was a repeat test.

### 3.6. Brain Tissue Staining

For each group of six rats, brain tissue was removed and placed in 4% paraformaldehyde for 24 h fixation, dehydrated, and embedded as wax blocks. Sections were cut to a thickness of 4 μm, dewaxed with xylene for 20 min, and hydrated with 100%, 95%, 80%, and 70% for 5 min for subsequent staining.

#### 3.6.1. Hematoxylin and Eosin (H&E) Staining

After dewaxing and hydration, the sections were stained with hematoxylin for 4 min, Hydrochloric acid alcohol differentiated for 5 s, and washed with running water for 5 min. Next, eosin staining was performed for 3 min, followed by gradient alcohol dehydration, xylene transparency, and sealing with neutral adhesive. All reagents were obtained from Solarbio Biotechnology Co., LTD. (Beijing, China). Tissue staining was observed under a light microscope (Nikon, Tokyo, Japan).

#### 3.6.2. Nissl’s Staining

We operated according to the manufacturer’s instructions (Beyotime Biotechnology, Shanghai, China). Sections were dewaxed and hydrated, placed in an oven at 60 °C, stained with toluidine blue stain for 30 min, dehydrated by gradient alcohol, and finally ylene transparency and sealing with neutral adhesive. Photographs were taken using an optical microscope (Nikon, Tokyo, Japan).

#### 3.6.3. Immunohistochemical Staining (IHC)

Brain tissue sections were dewaxed and placed in a repair kit filled with EDTA (PH 8.0) antigen repair solution for antigen repair in a microwave oven for 8 min at medium heat until boiling and then held for 8 min at a ceasefire and then turned to medium-low heat for 7 min. Sections were incubated sequentially with 3% H_2_O_2_ for 25 min and 3%BSA for 30 min. The primary antibodies were added dropwise, incubated overnight at 4 °C and incubated with secondary antibodies (HRP-labeled) at room temperature for 50 min. Finally, the cells were developed with DAB (Beyotime Biotechnology, Shanghai, China), re-stained with hematoxylin for 3 min, and photographed using a light microscope (Nikon, Tokyo, Japan). Immunofluorescence-positive areas were assessed using Image J analysis software (National Institutes of Health, Bethesda, MD, USA).

### 3.7. P-Tau217 ELISA Test 

A measurement of 100 mg of brain tissue was weighed, PBS was added, homogenized thoroughly, and centrifuged at 2000 r/min for 20 min, and the supernatant was collected. An ELISA kit (Jianglai Biologicals, Shanghai, China) was used for the assay according to the manufacturer’s instructions. The absorbance values of the samples were measured at 450 nm by a Scientific Multiskan GO enzyme marker (Thermo, Shanghai, China), and the concentration of each sample was obtained from the standard curve.

### 3.8. Absolute Quantification of Targeted Energy Metabolism

We weighed the brain tissue at 100 mg, added pre-cooled extraction solution 1000 μL, sonicated it in an ice bath, centrifuged it at 16,000× *g* for 30 min at 4 °C, and removed the supernatant. An equal amount of standard internal L-Glutamate_D5 was added to each sample and then vacuum-dried. For mass spectrometry detection, 80 μL of acetonitrile-water solution (1:1, *v*/*v*) was added for re-dissolution, and the supernatant was centrifuged at 16,000× *g* for 30 min at 4 °C, and the supernatant was taken into the sample for LC-MS/MS analysis. UHPLC was used for the separation by ShimadzuNexeraX2LC-30AD. QC samples were inserted in the sample queue for monitoring and evaluating the system’s stability as well as the experimental data’s reliability. Mass spectrometry was analyzed using a QTRAP5500 mass spectrometer in positive/negative ion mode. The MRM mode was used to detect the ion pairs to be measured. Data processing was performed using Multi Quant software to extract the chromatograms’ peak areas and retention times. Metabolite identification was performed using energy metabolite standards corrected for retention time. The standard internal L-Glutamate_D5 normalized the peak areas of metabolite-extracted ions for subsequent analysis. Metabolomics statistical analysis was performed using the MetaboAnalyst (http://www.metaboanalyst.ca/, accessed on 17 April 2022), the online statistical platform.

### 3.9. Cell Culture

Mouse hippocampal neuronal cells (HT-22 cells) were purchased from Shanghai QiDa Biotechnology Co., Ltd. (Shanghai, China) and maintained in Dulbecco’s Modified Eagle Medium (DMEM) in the presence of 10% fetal bovine serum and Pen/Strep antibiotics (GIBCO/Life Technologies, Grand Island, NY, USA) in a humidified incubator (5% CO_2_ at 37 °C).

### 3.10. MTT Assay for Cell Viability

HT-22 cells (1 × 10^4^ cells/well) were cultured in 96-well plates at 37 °C with 5% CO_2_ and exposed to 500 μM H_2_O_2_ for 12 h. Cells treated with a culture medium were used as a control only. After removing the supernatant of each well and washing it twice with PBS, 20 μL of 5 mg/mL MTT reagent (Biovision Inc., Milpitas CA, USA) was introduced and incubated for 4 h. After the supernatant had been removed and 200 μL of DMSO had been added, the wells were well mixed. The absorbance intensity was measured at 570 nm with the Scientific Multiskan GO enzyme marker (Thermo, Shanghai, China). Relative cell viability was expressed as a percentage relative to untreated control cells.

### 3.11. Mitochondrial Respiration Assay

The oxygen consumption rate (OCR) was measured with a Seahorse XF96 Extracellular Flux Analyzer (Seahorse Bioscience, North Billerica, MA, USA). HT-22 cells were inoculated at 1 × 10^5^ cells/well density into XFe96 cell culture microplates (Seahorse Bioscience). After treatment, the medium was changed to unbuffered DMEM (pH 7.4) supplemented with 1 mM pyruvate, 2 mM glutamine and 10 mM D-glucose 1 h before the assay. After the basal respiration was measured, oligomycin (1 μM), FCCP (1 μM), and rotenone (0.5 μM) with antimycin A (0.5 μM) were injected sequentially into each well. The OCR was recorded and normalized to 1000 cells per well, and the data were analyzed using Wave desktop software provided by Seahorse Bioscience.

### 3.12. Statistical Analysis

Values are expressed as the mean ± standard error of the mean (SEM). All statistics were analyzed using Sigma stat3.5 statistical analysis software, and comparisons between multiple groups were made using two-way ANOVA or one-way ANOVA. If the data were non-normally distributed or had uneven variances, ANOVA on Ranks was used for comparison. *p*-Values ≤ 0.05 were considered statistically significant. Graphs were prepared using GraphPad Prism 7.0 software.

## 4. Discussion

Rats subjected to 2-VO form a chronic hypoperfusion blood supply state, a common VD model [25]. The results of the MWM and passive avoidance test showed significant cognitive impairment in rats treated with 2-VO and that Gas and HBA had an ameliorating effect on the learning memory capacity of VD rats. The function of the neuronal cell was assessed by pathology, and the intervention of Gas and HBA reduced neuronal necrosis and edema. VD has many commonalities with AD, and VD can also induce Aβ and Tau protein deposition, further contributing to the development of dementia [26]. Previous studies have reported that Gas can exert neuroprotective effects by reducing the expression of Aβ and Tau proteins [22,27]. Our results follow expectations that Aβ1-42, p-tau396, and P-tau217 proteins were expressed in VD rat brain tissue and antagonized by Gas and HBA, most significantly at 50 mg/kg. The evaluation results showed no significant difference in the improvement of VD rats treated with 50 mg/kg of Gas and HBA. Previous studies have found that Gas has less brain exposure after entering the body, which is thought to be due to the rapid metabolism of Gas to HBA and poor permeability across the blood–brain barrier (BBB) [28]. The results of HBA metabolic distribution showed that the BBB had high permeability, but due to its small molecular weight, HBA would be rapidly metabolized to the body, and the content in the brain would also decrease [29]. Although Gas and HBA have limitations in their effects on the CNS, our results showed that both Gas and HBA had neuroprotective effects in VD rats. 

Glucose is the primary energy source of brain tissue, and neurons cannot produce and store glucose and perform normal aerobic metabolic activities when the brain is ischemic and hypoxic [30]. Normal energy metabolism is closely related to general cognitive responses, with the consequence that any deficiency in the energy metabolism process may lead to a decline in cognitive function [24]. Based on the clarification that Gas and HBA have ameliorative effects on VD, we performed the quantitative metabolomic analysis of energy metabolites to investigate the changes induced in the glycolytic pathway, TCA cycle, and pentose phosphate pathway after drug intervention. The results showed that the metabolite levels of glucose 6-phosphate, fructose 1,6-bisphosphate, and phosphoenolpyruvate were significantly increased in the brain tissue of VD rats in the glycolytic pathway. Phosphoenolpyruvate is a substrate for pyruvate kinase, which irreversibly converts phosphoenolpyruvate to pyruvate. Although the phosphoenolpyruvate concentration increased, the downstream pyruvate concentration did not change significantly, whereas the end-product of glycolysis, lactate, was significantly increased. We consider that the increase in upstream and downstream metabolites of glycolysis indicates that many substrates have flowed into the pathway. Additionally, because of the lack of oxygen, the entry of NADH into the respiratory chain is blocked, and the concentration increases. When pyruvate is decarboxylated to acetyl-CoA by oxidation, this enters the TCA cycle [31]. The TCA cycle is the main metabolic pathway for ATP production by the electron transport chain [32] as metabolites of the TCA cycle, citric acid, oxaloacetic acid, and acetyl-CoA, were significantly down-regulated in VD. The analysis results also showed that the levels of ATP and AMP in the brain tissue of VD rats were significantly down-regulated. The above results suggest that brain energy requirements are substantially reduced after ischemia and hypoxia, but glycolysis persists, indicating that complex energy metabolism changes may affect neuronal responsiveness to ischemia and hypoxia [33].

Brain tissue metabolites in VD rats changed after Gas and HBA intervention, where Gas and HBA reduced glucose 6-phosphate content, but intermediate metabolites of the glycolytic pathway, such as 2-phosphoglycerate and 3-phosphoglycerate, were elevated, while lactate, the end-product of glycolysis, was not significantly changed. It has been suggested that although the brain requires large amounts of energy, neurons with truncated glycolytic pathways may serve as a protective mechanism after brain injury [34]. We suggest that Gas and HBA may influence the flux of the glycolytic pathway. In addition, in the analysis of metabolites in the TCA cycle, we found that although there were no significant changes in key metabolites in the brain tissue of VD rats, Gas and HBA significantly increased the content of critical metabolites such as citric acid, succinic acid and fumaric acid in the TCA cycle. We significantly increased the content of ATP, and we consider that Gas and HBA rescued the energy of VD rats which had to some extent been depleted.

On the other hand, Gas increases the content of ribose 5-phosphate involving changes in the pentose phosphate pathway. The pentose phosphate pathway produces ribose 5-phosphate for nucleotide synthesis, which controls the metabolic synthesis and redox homeostasis [35]. Some findings suggest that enhancing the pentose phosphate pathway is a potential target for treating ischemic brain injury [36].

Energy metabolism occurs mainly in mitochondria [37], and mitochondrial dysfunction has been well documented as an early event in dementia [38]. The assessment of the mitochondrial function is crucial for understanding energy metabolism-related diseases and the development of corresponding drugs, and mitochondrial respiration is an essential indicator for assessing mitochondrial function. To simulate the in vivo situation in a VD model, we used H_2_O_2_ to stimulate HT-22 cells leading to oxidative stress damage [39], and H_2_O_2_-induced oxidative damage in neuronal cells can lead to cellular mitochondrial damage, resulting in dysregulated mitochondrial respiration and ATP production [40]. Mitochondrial OCR is one of the most important indicators for assessing mitochondrial respiration [41]. Our results showed a significant decrease in cell viability after H_2_O_2_ treatment, and Gas and HBA increased the cell viability of HT-22 cells in a dose-dependent manner. Measuring real-time mitochondrial respiration in HT-22 cells using Seahorse XF extracellular flux analysis showed that H_2_O_2_ significantly inhibited the mitochondrial respiratory capacity of HT-22 cells, and Gas and HBA ameliorated this inhibitory effect, including increasing neuronal ATP production. Previous studies have shown that repair of mitochondrial dysfunction prevents hippocampal neuronal damage [42] and that maintenance of mitochondrial function by Gas and HBA may underlie the protection of HT-22 cells from oxidative damage.

These data suggest that Gas and HBA enhance neuronal energy metabolism and improve neuronal mitochondrial function.

## 5. Conclusions

VD rats showed significant impairment in learning memory function, prominent accumulation of Aβ and Tau proteins, and disturbances in brain energy metabolism and mitochondrial dysfunction. Gas and HBA can exert neuroprotective effects by improving learning memory abilities and neuronal damage, reducing Aβ deposition and Tau protein phosphorylation, improving brain energy metabolism disorders in rats, and reducing mitochondrial dysfunction induced by H_2_O_2_ oxidative damage to cells.

## Data Availability

All data used in this study are available from the corresponding authors upon reasonable request.

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
