# Peer review of "Gastrodin and Gastrodigenin Improve Energy Metabolism Disorders and Mitochondrial Dysfunction to Antagonize Vascular Dementia"

_molecules, 2023, doi:10.3390/molecules28062598_

Round 1
Reviewer 1 Report
This work reported the possible neuroprotective effects of Gastrodin and Gastrodigenin on vascular dementia, including regulation of energy metabolism and mitochondrial function. This work is kind of interesting, however, there are some flaws that need to be addressed.
1. Since the role of gastrodia-related bioactive substances in vascular dementia is now well understood, the aim of the author's study is to reveal new mechanisms. Therefore, the title of the article should be changed to a similar description below "Gastrodin and Gastrodigenin Improve Energy Metabolism Disorders And Mitochondrial Dysfunction to Antagonize Vascular Dementia"
2. The results of the above two mechanisms should be described centrally in the abstract, and should not be scattered; Also, there seems to be a lot of extra space in the abstract, please check
3. The lack of spaces between numbers and letters in methods, main text and figure legends is a serious descriptive error. Authors should check it carefully and make corrections.
4. Why is the passive avoidance test performed after the Morris water maze test?
5. The authors need to revise many details, for example, the ordinate titles of Figure 1C and D were the same meaning and did not match the legend. Anyway, I can't understand exactly what each represents from Figure 1C-F, the author should give the correct ordinate titles. And, I consider “seaech” a misspelling.
6. Tissue-based quantitative detection of Aβ and p-tau is very important, because the results of IHC are very subjective. But I only find quantitative results of p-tau in brain, the authors need to provide the western blot or ELISA results of Aβ.
7. The abscissa number is not visible in Figure 4C
8. Figure7 Legend B shows only energy metabolism regulation, whether mitochondrial function regulation should also be included? In addition, this schematic should remove substances that are not examined, such as lactate.
9. For “The evaluation results showed that there was no significant difference in the improvement of VD rats treated with 50mg/kg of Gas and HBA. However, HBA 25 mg/kg was less effective than Gas 25 mg/kg”, what kind of assessment does this sentence refer to? It was then explained that Gas is rapidly metabolized to act as HBA, so why is HBA less effective than Gas?
Reviewer 2 Report
Wu est al. described a study entitled Gastrodin and Gastrodigenin Antagonize Vascular Dementia By Improving Energy Metabolism Disorders And Mitochondrial Dysfunction. The authors concluded that both compounds exerted neuroprotective effects against vascular dementia by regulating energy metabolism and mitochondrial function. The experimental design is well. The conclusion supported the results. Overall the manuscript is appropiate for publication.
Author Response
Dear reviewer, Thank you so much for handling the review of our manuscript, and your favorable consideration of our manuscript is greatly appreciated.
Reviewer 3 Report
This manuscript by Wu and colleagues examined the effects of Gastrodin and Gastrodigenin on vascular dementia (VD) in rats. Bilateral common carotid artery ligation is used to model VD in rats, and Morris water maze and passive avoidance test were used to assess cognitive function. The animal model for vascular dementia is appropriate and the experimental designs and data qualities are generally sound. The statistical analysis is appropriate when it is used.
Major concerns:
1. Whereas all other assays are quantitative, assessment of neuronal damage in 3.2 is not. Authors should use stereology to quantify the effects of gastrodin and gastrodigenin on neurodegeneration if they have access to equipment and software for stereology. If not, they should consider removing this section from the results.
2. Justification for the use of HT-22 cells needs to be clarified. Seahorse assay can be performed on hippocampal tissues. Why authors switch to a not-well-characterized cell line?
Minor:
1. Line 60-61: change “is the main medicinal part, which” to “is a Chinese herbal medicine that”
2. Line 64: insert “are active components of rhizome of Gastrodia Elata and” before “of interest”
3. Methods section 2.7 and 2.8 need to be edited for English grammar and usage.
Round 2
Reviewer 1 Report
The authors have taken my comments into account and have adjusted the manuscript accordingly.